# Patients’ Perspectives on Determinants Avoidable Hospitalizations: Development and Validation of a Questionnaire

**DOI:** 10.3390/ijerph19053138

**Published:** 2022-03-07

**Authors:** João Sarmento, Margarida Siopa, Rodrigo Feteira-Santos, Sílvia Lopes, Sónia Dias, António Sousa Guerreiro, António Panarra, Paula Nascimento, Afonso Rodrigues, Ana Catarina Rodrigues, João Victor Rocha, Rui Santana

**Affiliations:** 1Public Health Research Center, NOVA National School of Public Health, Universidade NOVA de Lisboa, 1600-560 Lisboa, Portugal; joaoccsarmento@gmail.com (J.S.); silvia.lopes@ensp.unl.pt (S.L.); sonia.dias@ensp.unl.pt (S.D.); ruisantana@ensp.unl.pt (R.S.); 2NOVA National School of Public Health, Universidade NOVA de Lisboa, 1600-560 Lisboa, Portugal; margarida.siopa@ensp.unl.pt; 3Instituto de Saúde Ambiental, Faculdade de Medicina, Universidade de Lisboa, 1649-028 Lisboa, Portugal; rodrigosantos@medicina.ulisboa.pt; 4Área Disciplinar Autónoma de Bioestatística, Faculdade de Medicina, Universidade de Lisboa, 1649-028 Lisboa, Portugal; 5Comprehensive Health Research Center, 1600-560 Lisboa, Portugal; 6Serviço de Medicina 4, Hospital de Santa Marta, Centro Hospitalar Universitário Lisboa Central, 1649-028 Lisboa, Portugal; sousaguerreiro@hotmail.com (A.S.G.); med.paulaoliveira@gmail.com (P.N.); j.afonso.sr@gmail.com (A.R.); 7Serviço de Medicina 7.2, Hospital Curry Cabral, Centro Hospitalar Universitário Lisboa Central, 1050-099 Lisboa, Portugal; acpanarra@chlc.min-saude.pt (A.P.); anacjrodrigues@gmail.com (A.C.R.)

**Keywords:** ambulatory care sensitive conditions, patients’ perspective, determinants

## Abstract

Ambulatory care sensitive conditions (ACSC) can be avoided through effective care in the ambulatory setting. Patients are the most qualified individuals to express the social and individual contexts of their own experience. Thus, understanding why potentially preventable hospitalizations occur is important to develop patient-centred policies or interventions that may reduce them. This study aims to develop and validate a questionnaire to capture the patients’ perspective on the causes of the hospitalizations for ACSC. The development of a new questionnaire involved four phases: a literature review, face validity, pre-test, and validation. We conducted a three-step face validity verification to confirm the relevance of the identified determinants and to collect determinants not previously identified by interviewing healthcare providers, representatives of patients’ associations, and patients. Determinants were identified through the literature review predominantly in the “Healthcare Access”, “Disease self-management”, and “Social Support” domains. The validated resulting questionnaire comprises 25 questions, distributed by two dimensions (individual/contextual) covering seven domains and 20 determinants of ACSC hospitalization. Currently, there are no validated instruments as comprehensive and easy to use as the one described in this paper. This questionnaire should provide a base for further language/context validations.

## 1. Introduction

Patients are the most qualified individuals to express the social and individual contexts of their own experience. They should be an important source of information, providing a more holistic and long-term view of the factors that contribute to hospitalizations. If patient-centred care is advocated, the patients’ point of view needs to be included in the research process to centre outcomes to their interests and values [1,2]. This information is crucial to understanding the reasons for hospitalizations and the specific challenges that need to be addressed in the design of ambulatory care [3].

Ambulatory care sensitive conditions (ACSC) are defined as health conditions for which hospitalization or emergency care can be avoided through addressing these conditions effectively in the ambulatory setting [4,5,6]. These conditions are frequently used as a proxy to measure potentially preventable hospitalizations [7]. Three types of influencing factors were proposed by Nedel et al. [7], namely: geographical characteristics; sociodemographic characteristics; and model of care. The first category includes variables such as lower populational density and isolation [8], or hospital proximity [9], which tend to aggravate ACSC hospitalizations. Lower-income, lower education, higher age, or higher unemployment rates are examples of sociodemographic characteristics associated with higher rates of ACSC hospitalizations. The number of comorbidities is a further example of a significant risk factor [10], which increases the probability of hospitalization with each chronic condition or body system affected. Finally, the model of care influences ACSC hospitalizations. The availability of healthcare providers has an impact, with lower rates where there is higher availability [8,11], but organizational characteristics such as continuity of care also play an essential role.

In Portugal, ACSC hospitalizations are estimated to account for 12% of hospitalizations, with a projected financial impact of up to 350 million EUR/year [12]. The most frequent were pneumonia, chronic obstructive pulmonary disease, heart failure, hypertensive heart disease, and urinary tract infections [13,14].

Thus, understanding why potentially preventable hospitalizations occur is crucial to developing policies and interventions to reduce them. In addition, many authors consider avoiding unnecessary hospitalizations a relevant indicator of the quality of ambulatory care and the efficiency of the health system [6,15].

Most evidence and information on ACSC hospitalizations herein mentioned is obtained through hospitals’ administrative discharge databases. The perspective of healthcare professionals on the causes of avoidable hospitalizations or emergency care has also been studied [16,17], which allows us to draw a rough map of the determinants of ACSC hospitalizations at the population level [3]. However, little research captured the patient’s perspective on the leading causes of ACSC hospitalizations [1] and what could have been done to avoid it. One study in Australia conducted semi-structured interviews with patients and health professionals to identify factors associated with potentially avoidable hospitalizations in a rural context [18]. Authors identified complex and interrelated factors associated with potentially avoidable hospitalizations, classified into five themes: General Practitioner involvement, individual patient factors, the influence of the rural locality, medication awareness, and health service access. In a study in Brazil, patients hospitalized for avoidable conditions were asked to answer a questionnaire with clinical, socio-economic, and demographic characteristics, as well as primary health care assessment items [19]. Patients gave low grades to access, family focus, and community orientation, while longitudinality was identified as a relevant dimension for continued care, crucial for avoiding such hospitalizations. Both studies agree that promoting access to effective health services is necessary to the challenge of avoiding hospitalizations.

Analysing the patients’ perspective can provide information beyond the known provider-centred data [3]. However, to the best of our knowledge, there is no published questionnaire to capture the patients’ perspective about the causes of ACSC hospitalizations, be used across different populations to identify individual determinants, and allow the comparability of results and effective design of integrated care interventions.

This study aims to develop and validate a questionnaire to capture the patients’ perspectives on the causes of the hospitalizations for ACSC.

## 2. Materials and Methods

This is a design and validation study with multiple phases of data collection to develop a new questionnaire intended to capture the patients’ perspectives on the causes of the ACSC hospitalizations. Figure 1 presents the four phases that this study involved, namely a literature review (phase 1), face validity (phase 2), a pre-test (phase 3), and validation (phase 4). 

### 2.1. Phase 1: Literature Review

The purpose of this literature review was to map relevant research and evidence on reasons or causes of ACSC hospitalizations according to the patients’ perspective and also to confirm if other instruments had already been developed to assess the same construct [20].

Research of relevant literature on the theme was performed between October and November 2017 on the indexed databases PubMed and Medline, using the keywords: patients’ perspective, reasons/causes, ACSC, and preventable/avoidable admission/hospitalization. No criterion for the inclusion was considered regarding the date of publication, and papers were selected, after reading the title and abstract, if the focus was the reasons or causes of potentially preventable hospitalizations in the patients’ perspective. Papers using the family members or health professionals’ perspective as a proxy of patients’ perspective, or exploring administrative databases for the causes of ACSC, were also considered. References of selected papers were also screened to identify further relevant research. Moreover, grey literature related to patients’ perspectives on causes of ACSC hospitalizations was also considered.

The several items (representing reasons or causes of ACSC) identified in the selected papers were classified into determinants and domains according to the Sentell et al. [3] conceptual model. It was considered through a team consensus as being the most complete model of structured items and domains. Moreover, Sentell and colleagues’ conceptual model explored the patients’ perspective through interviews with open-ended questions, resulting in a broader perspective [3]. Finally, a further layer of aggregation was added to distinguish “Individual” and “Contextual” domains.

### 2.2. Phase 2: Face Validity

To further strengthen the findings of the literature review, we conducted a three-step face validity verification (phase 2a, 2b, and 2c) to confirm the relevance of the determinants identified through the literature review and to collect determinants not previously identified.

#### 2.2.1. Phase 2a: Semi-Structured Interviews (Experts, Representatives of Patients’ Associations, and Patients)

##### Experts

We invited healthcare professionals from the primary health care (PHC) and hospital settings. Healthcare professionals were asked, in their opinion, which elements the patients would identify as determinants of ACSC hospitalizations. The interviews followed a semi-structured approach. Firstly, the theme and aim of the interview were introduced. Next, we explained the definition of ACSC and referred to examples of these conditions. We then highlighted the specific purpose of using the patients’ perspective on the reasons or causes for ACSC hospitalizations. Secondly, we asked interviewees to identify reasons for ACSC hospitalization based on their experience. Thirdly, we explored the previously identified domains that each interviewee did not mention. Finally, we asked for suggestions on developing the questionnaire for the determinates they had mentioned. The identified items were categorized into determinants and domains using the same methodology followed during the literature review (Table 1).

##### Patients’ Representatives

We also conducted semi-structured interviews with representatives of patients’ associations with conditions related to ACSC hospitalizations. The methodology was identical. Once again, after those interviews, the items referred were categorized into determinants and domains using the same methodology as followed during the literature review.

Table 2 describes the patients’ associations whose representatives were interviewed.

##### Patients

A semi-structured guide was also developed to interview the patients with open-ended questions based on the work of Sentell et al. and Glasby et al. [1,3]. The first six questions from Sentell et al.’s questionnaire [3] were selected through a team consensus. Then, these questions were translated and added two supplemental questions, based on the items mentioned by healthcare professionals and representatives of patient’s associations’ and on Glasby et al. [1] (interview form available from authors upon request). 

The patients’ interviews took place in two internal medicine wards of Central Lisbon University Hospital Centre (CHULC). The medical team was asked to verify the admissions in order to identify potentially eligible patients with the following inclusion criteria:Age over 18 years old;Being conscious, clinically stable, and capable of responding (evaluated by a clinician);Hospitalized for at least three days on an Internal Medicine Ward of CHULC;Unplanned hospitalization;Having a most responsible cause of admission meeting one of the following: chronic obstructive pulmonary disease (COPD), diabetes, asthma, urinary tract infection, congestive heart failure, or hypertension. The latter are considered ACSC hospitalizations according to the Prevention Quality Indicators (PQI) developed by the Agency for Healthcare Research and Quality [21].

The medical team invited the patient to participate and notified the research team in case of acceptance. All interested participants provided written informed consent. Recruitment was stopped when thematic saturation on patient-reported items was reached. 

The inputs obtained through the literature review and the semi-structured interviews resulted in a first draft questionnaire (Table 3). The determinants mentioned at least five times were included in the questionnaire. To translate the identified determinants into questions/scales, we again researched indexed databases and grey literature [22,23,24,25,26,27,28,29,30,31,32,33,34,35,36,37,38,39].

#### 2.2.2. Phase 2b: Experts Validity

An expert panel comprised ten health care professionals from areas such as primary health care, internal medicine, health management, and health promotion research. The selection was based on their knowledge and expertise and recruited by personal contact. In addition, experts evaluated the draft questionnaire on content such as absences, redundancies, and ambiguous or confusing questions, adequacy of the Likert scale as the most proper answering format, and the questionnaire as a whole. 

Relevant measures of patient characteristics were also considered to allow for posterior construct validity analysis, as well as questions on sociodemographic characteristics.

#### 2.2.3. Phase 2c: Cognitive Interviews

We conducted cognitive interviews with patients hospitalized for ACSC to determine the comprehension and interpretation of the questions and terms [40,41]. 

The interviewer used techniques such as “Think aloud”, “Probing”, and “Paraphrasing” in a way in which unexpected problems emerged with some items and specific terms used on the questions [40,41].

### 2.3. Phase 3: Pre-Test

The medical team followed the same criteria used on the patients’ semi-structured interviews to recruit potentially eligible patients to be included in the pre-test phase. All interested participants provided written informed consent after being informed of the purpose of the study. An interviewer administered the questionnaire with previous experience in a similar data collection method. 

### 2.4. Phase 4: Validation

The sample size for this phase was defined based on the healthcare activity of CHULC’s wards, where the interviews would be conducted. The number of beds and admissions observed in internal medicine wards in the CHULC per year (according to the activity report for 2015), and the frequency of admissions by the most common ACSC in Portugal (COPD, diabetes, asthma, urinary tract infection, congestive heart failure, and hypertension) were used to calculate the number of potential participants during the period defined for the data collection. Therefore, it was expected that between 140 and 210 patients could be invited to participate.

Once again, the medical team of CHULC was responsible for recruiting potentially eligible patients, following the same criteria used on the patients’ semi-structured interviews. All interested participants provided written informed consent after being informed of the purpose of the study. All interviews were conducted by the same interviewer, who had already administered the questionnaire during the pre-test phase.

The questionnaire was developed and validated in Portuguese. The final questionnaire was translated into English for a better comprehension of this process (Appendix A).

## 3. Results

### 3.1. Phase 1 (Literature Review) and 2 (Face Validity)

Nine papers were identified through the literature review. Four papers used the patients’ perspective [1,2,3,17], two the professionals’ perspective [4,16], and one used a population-based approach using an administrative database to predict the individual risk of ACSC hospitalization [5]. Three reviewed evidence about determinants of hospitalizations [7,31,39].

In total, seven healthcare professionals and six representatives of patients’ associations with conditions related to ACSC hospitalizations were interviewed.

Moreover, twenty-two patients admitted with an ACSC were interviewed.

The identified determinants were predominantly categorized in the “Healthcare Access”, “Disease self-management”, and “Social Support” domains. Each of the formerly described groups of interviewees (healthcare professionals/patients’ representatives/patients) mentioned different determinants and with different emphases. All sources coincided in the most frequently mentioned domain of determinants: “Healthcare Access”. The second most frequently mentioned domain was “Disease self-management”, except in healthcare professionals, who emphasized the “Social Support” domain. The following more frequently mentioned domains differed. Patients identified “Health Status”, Representatives of patients’ associations stated “Lifestyle”, while Healthcare professionals mentioned “Disease self-management”. (Table 3).

During phase 2b, experts evaluated the draft questionnaire, and their inputs led to the questionnaire’s evolution, as listed in Table 4. For the individual dimension, we developed twelve questions on the determinants of ACSC hospitalizations. For the contextual dimension, the team developed thirteen questions adapted from several tools [25,36,38].

Three whole scales validated in Portuguese(mental health [26,27], health literacy [28], and social support [29,30,32])and one translated by the team)disease management self-efficacy [24])—were considered to include in the draft questionnaire to allow for posterior construct validity. Finally, ten questions on sociodemographic characteristics were added to further characterize the patients [32,33]. Other characteristics were not measured to keep the length of the interview feasible. The selection of the characteristic to be measured took into account the team experience, the reviewed literature, and the experts’ opinion.

Four cognitive interviews were performed during phase 2c with patients hospitalized for ACSC, identified by the CHULC medical team. The analysis of the interviews characterized the direct representation of the determinants from the patients and allowed slight modifications to the wording when indicated by the patient to improve understanding.

### 3.2. Phase 3 (Pre-Test)

Fourteen patients hospitalized for ACSC at two internal medicine wards of CHULC were surveyed in this phase from September to October 2018. This phase revealed that patients better understood the questions when formulated in the negative. The analysis of the answers obtained also led to modifying the wording of six items to improve understanding and achieve the final questionnaire to be used on the field study.

### 3.3. Phase 4 (Validation)

From November 2018 to May 2019, 197 patients hospitalized for ACSC at two internal medicine wards of CHULC were selected to participate in the study. A total of 132 individuals accepted to participate in the study: 123 completed the survey, 7 only completed the individual, contextual, and sociodemographic parts, and 2 were excluded due to incomplete surveys (Figure 2).

The internal consistency of the questionnaire was evaluated through the survey results. For the analysis, we used the coefficient alpha, also known as Cronbach’s alpha, with higher values indicating a stronger interrelation with one another (Table 5). The individual dimension of the questionnaire yielded an alpha of 0.647, and the contextual dimension yielded an alpha of 0.753.

Then, the construct validity of the survey was evaluated by estimating the association between the survey questions and other measurable variables. Following the original questionnaire we also applied pre-existing tools to evaluate the patients’ mental health status [26,27], health literacy [28], social support [29,30,32], and disease management self-efficacy [24]. The analysis of patterns of associations is described in Table 6.

## 4. Discussion

This paper described the development of a questionnaire capable of capturing the patients’ perspectives on the causes for hospitalizations for ACSC. This questionnaire, presented in Appendix A, comprises eight domains of individual and contextual dimensions. Its comprehensiveness, validity, and adequacy were validated through interviews with 130 patients in Portugal. Usually, the available research on ACSC is mostly population-based, through the analysis of hospitals’ administrative discharge databases; these data lack information of key factors that play critical roles in hospitalizations for ACSC. Furthermore, other information, such as behaviours (diet, exercise, and consumption of alcohol, drugs, and tobacco) or mental health (feeling anxiety or sadness), that could be present in administrative data (although possibly underreported) are usually not included in ACSC analysis. Therefore, eliciting the perspective of the hospitalized patients brings valuable insights that are usually overlooked in studies on ACSC.

Few studies have analysed patients’ perspectives on what could have contributed to the ACSC hospitalizations. A study in Hawaii employed interviews with patients to develop a model to understand pathways to hospitalizations, consisting of three categories of factors: immediate (urgent reason for hospitalization), precipitating (the reason that led to the urgency matter), and underlying factors (relating to challenges or circumstances that lead to precipitating factors) [3]. Factors found as relevant included homelessness, limited patient knowledge, and poor health care system coordination/communication, which may help explain why clinical-focused disease-management interventions in preventing hospitalizations may not be as successful as intended [3]. A study on patients’ perspectives in Australia has found that lack of social support was an important factor possibly associated with the hospitalization for ACSC [42]. Authors argued that, although patients did not connect lack of social support to the hospitalization, or even did not perceive their admission to be preventable, the factors identified may have contributed to the hospitalization nonetheless [42]. A study in Brazil performed structured interviews with patients using a primary care assessment tool [43], with participants reporting difficulties in access (regarding making timely appointments and consultation times) and excessive focus of health professionals on the disease (to the detriment of social determining conditions) as factors associated with the hospitalization for ACSC [19]. These studies agree that analysing patients’ perspectives can provide key insights on reasons of hospitalizations for ACSC, which is crucial to design effective interventions to reduce these events.

The aforementioned papers used interviews to collect data. This method is somewhat time and resource consuming, both on the collection and processing of data, and therefore has limited applicability. To the best of our knowledge, there are no studies of validated instruments to capture patients’ perspectives on ACSC as comprehensive and easy to use as the one described in this paper, which is the main strength of this questionnaire. 

Patients’ experiences on ACSC are highly relevant to understanding what leads to these events; these are usually neglected compared to other perspectives obtained by ecological research approaches. The questionnaire developed can help fill this gap in ACSC research and provide more information for the decision-making process, both on strengthening care provided at the PHC and hospital levels and how to involve other sectors to tackle factors outside the clinical scope. In addition, this questionnaire can prompt interventions resulting from the patients’ opinions. For example, these interventions can encompass activities such as adapting social care pathways and work schedules (social support domain); remotely controlling adherence to medication and measuring blood glucose and pressure through apps (disease self-management domain), strengthening literacy (health literacy domain), developing health promotion group activities (lifestyle domain) and improving referral to mental health services (health status domain). There may exist individual causes that can be adapted by case management, as well as patterns for a group of patients that would require community interventions.

There are practical questions about using such a questionnaire that should be addressed, including how often to apply it to patients, who would be the professional responsible for it, how the data are stored and treated, and how it can be integrated into other sources of information. It is crucial to move from theory to practical interventions to effectively solve these questions, optimizing this instrument’s potential in avoiding hospitalizations and improving care.

Although the use of this questionnaire in different contexts is encouraged, the limitation of the validation process must be acknowledged, as the participants involved (experts and patients) were Portuguese. The questionnaire included questions on sociodemographic characteristics of patients, to allow for construct validity analysis. Nonetheless, the external validity regarding sociodemographic characteristics for different populations was not assessed. Therefore, the questionnaire might not be entirely adequate for application in all contexts due to either cultural characteristics or health system organization. However, this questionnaire should provide a base for further language/context validations.

The sample size used in this validation process was established according to the number of potential patients treated in CHULC internal medicine wards during data collection. A review of publications on newly developed patient reported outcomes measures stated that clear recommendations on the sample size definition for validating these scales still need to be developed [44]. However, the number of participants in this study’s validation stage matched the size range observed in other studies to validate similar scales to assess patients’ perspectives [44].

The test–retest method is frequently used to validate the reliability of questionnaires. However, this method was not used in this research once the application of the questionnaire caused the patients to reflect about the avoidability of the admission and furthermore explores potential specific causes of admission that we once hypothesized might bias future answers by the same individual. We used Cronbach’s alpha coefficient to evaluate the internal consistency of the questionnaire. The results are acceptable, according to Tavakol and Dennick [45]. The lower coefficient in the individual dimension was expected once the purpose of the questionnaire was to capture “singularities” of the individual that may explain the reason for admission. Furthermore, this dimension is broader than the contextual dimension. This means that if patients frequently singled out a specific reason for admission, the coefficient tends to drop.

### Construct Validity Discussion

There are significant associations between the questionnaire answers and patient characteristics that are logical and therefore support the construct validity of the questionnaire. The correlations between the Manage Disease in General Scale and individual and contextual dimensions are both significant, although weak, and have logical opposing slopes. This means that individuals with higher capacities to manage disease often tend to identify causes for ACSC hospitalization related to the context than with themselves. This idea is further supported by the positive correlation between Health Literacy Index and the context dimension. The negative correlation of the mental health inventory score with the individual dimension and the Health status domain, where mental health status is included as a determinant, means that individuals with poorer mental health identify more frequently individual factors and, more precisely, their health status as causes for hospitalization.

The authors propose this questionnaire as a valuable tool that can provide a better understanding of individual and contextual factors that lead to hospitalizations for ACSC. It can help identify shortcomings in the quality of the health care provided and help identify patients who had a higher risk of being hospitalized, thus who should be followed up more closely in the outpatient setting. As these events are potentially preventable, their reduction would bring positive results for health systems and populations in terms of reduced costs and better health quality.

## 5. Conclusions

Most evidence on ACSC hospitalizations is obtained through hospitals’ administrative discharge databases. Little research captures the patients’ perspective on the leading causes of ACSC hospitalizations. No validated questionnaire to capture the patients’ perspective has been previously published. Our literature review identified several determinants of ACSC hospitalizations aggregated into domains and dimensions. Other determinants were identified by interviewing health care professionals and, most importantly, patients. Based on the identified determinants, we developed a comprehensive set of questions that evaluate to what degree specific determinants contributed to an ongoing ACSC hospitalization through the patients’ perspective. The questionnaire comprises 25 questions, distributed by two dimensions (individual/contextual) covering seven domains and 20 determinants of ACSC hospitalizations. The questionnaire’s internal consistency was evaluated with an acceptable Cronbach’s alpha coefficient, following its application to 130 patients during admission for ACSC. The construct validity was verified by logical associations between some patients’ characteristics and their answers to the questionnaire.

The developed questionnaire proved valid to reveal individual and contextual factors associated with ACSC hospitalizations that are usually missing in the administrative database analyses of these events. This information is a valuable aid that can be used in practical matters to guide the development of activities and interventions meant to improve care. The resulting reduction in hospitalizations for ACSC would indicate an improvement in the patients’ quality of life and a more equitable and effective health system.

## Figures and Tables

**Figure 1 ijerph-19-03138-f001:**
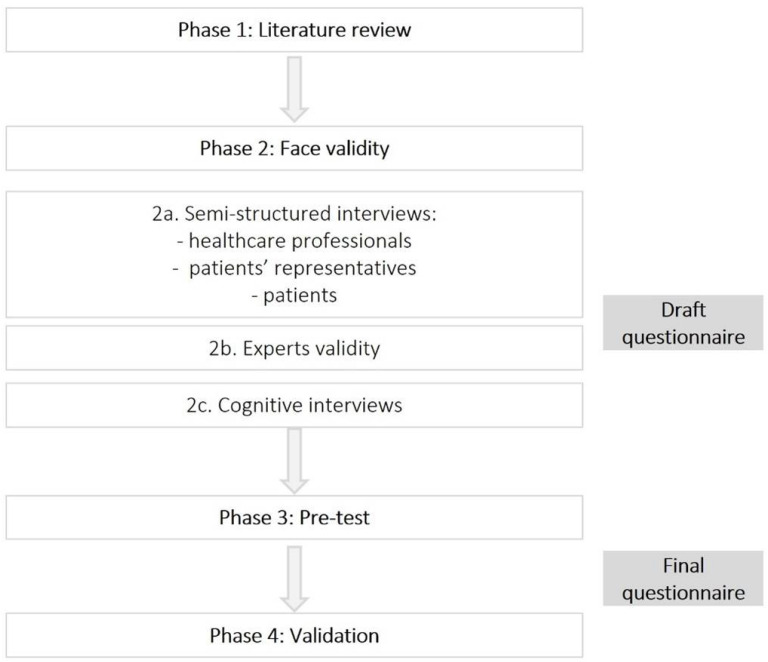
Phases of the study.

**Figure 2 ijerph-19-03138-f002:**
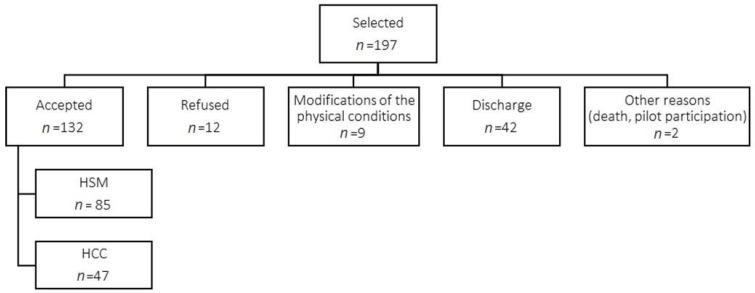
Final Sample Size. Legend: HSM—Hospital Santa Marta; HCC—Hospital Curry Cabral.

**Table 1 ijerph-19-03138-t001:** Dimensions and domains of determinants of ACSC hospitalizations.

Dimension	Domain	Determinants
Individual	Individual characteristics	Age; sex; marital status; ethnicity; income; employment
	Social support	Family support; friend support; patients’ association support; social services
	Health literacy	Knowledge about health; knowledge about their condition; knowledge about signs or symptoms, knowledge of the therapeutic plan
	Health status	Multimorbidity; mental health; substance abuse; functional status
	Disease self-management	Therapeutic plan adherence; medication adherence; condition signs control; treatment maintenance Busy; forgot; responsibilities; health as a second priority, denial of health condition Avoidance to accept the disease; avoidance to accept stress with a condition; avoidance to ask for help
	Lifestyle	Smoking habits; dietary habits; physical activity; hydration
Contextual	Environment characteristics	Rurality; city; pollution; social values; isolation
	Healthcare access	Availability of services; accessibility of services; health costs; quality; coordination of services; appropriateness of services; relation with healthcare professionals

**Table 2 ijerph-19-03138-t002:** Patients’ associations involved in semi-structured interviews.

Patients’ Associations	English Translation	Description
RESPIRA—Associação Portuguesa de Pessoas com DPOC e outras Doenças Respiratórias Crónicas	Portuguese Association of People with COPD and other Chronic Respiratory Diseases	To represent people with chronic obstructive pulmonary disease and other chronic respiratory diseases
APD—Associação Portuguesa da Doença Inflamatória do Intestino	Portuguese Association of Inflammatory Bowel Disease	To represent people with inflammatory bowel disease
GAT—Grupo de Ativistas em Tratamentos	Treatment Activists Group	To represent people with HIV/AIDS
APA—Associação Portuguesa de Asmáticos	Portuguese Association of Asthmatics	To represent people with asthma
AAGI-ID—Associação dos Amigos da Grande Idade—Inovação e Desenvolvimento	Association of Friends of the Great Age—Innovation and Development	To aggregate people working in the ageing sector
MAIS PARTICIPAÇÃO, melhor saúde	MORE PARTICIPATION, better health	To promote participation and empowerment of representatives of people with or without disease

Notes: The first column provides the Portuguese names of associations consulted in this study. The second column provides an English translation for the name of each patients’ association.

**Table 3 ijerph-19-03138-t003:** Number of individual items identified in semi-structured interviews according to each domain of the determinants of ACSC hospitalizations.

Dimension	Domain	Literature Review	Healthcare Professionals	Representatives of Patients’ Associations	Patients	Total
Contextual	Healthcare access	32	55	51	26	164
Individual	Disease self-management	19	16	20	18	73
Individual	Social support	19	24	10	3	56
Individual	Health status	13	9	5	15	42
Individual	Lifestyle	11	12	12	5	40
Individual	Health literacy	10	13	11	6	40
Individual	Individual characteristics	12	5	0	0	17
Contextual	Environment characteristics	3	7	3	2	15

**Table 4 ijerph-19-03138-t004:** Evolution of the list of determinants used on the questionnaire.

Draft Questionnaire		Final Questionnaire	
Individual	Knowledge about signs or symptoms	Individual	Informal care
	Knowledge of therapeutic plan		Formal care
	Informal care		Knowledge about signs or symptoms
	Formal care		Knowledge of therapeutic plan
	Help to maintain treatment		Help to maintain treatment
	Isolation		Incapacity to disease self-management
	Disease self-management		Therapeutic plan adherence
	Therapeutic plan adherence		Lifestyle
	Dietary habits		Multimorbidity
	Cultural barriers		Isolation
	Multimorbidity		Mental health
	Mental health		Denial of health condition
		Scales	Informal care
			Health literacy
			Disease self-management
			Mental health
Contextual	Coordination of services	Contextual	Coordination of services
	Adequacy of healthcare services		Adequacy of healthcare services
			Difficulty in consulting a practitioner
	Relation between patient and practitioner		Relation between patient and practitioner
	Healthcare professionals’ behaviour		The complexity of healthcare services
	The complexity of healthcare services		Communication between patient and practitioner
	Communication between patient and practitioner		Healthcare professionals’ behaviour
			Early disease diagnostic

			Cultural barriers
	Quality of life		Sociodemographic characteristics
	Sociodemographic characteristics		

**Table 5 ijerph-19-03138-t005:** Cronbach’s alfa.

Dimension	Domain	Determinant	Question	Number of Items	Alfa
Individual	Social support	Informal care	1.1	4	0.647
		Formal care	1.2		
		Help to maintain treatment	1.5		
		Isolation	1.10		
	Disease self-management	Incapacity to disease self-management	1.6	3	
		Therapeutic plan adherence	1.7		
		Denial of health condition	1.12		
	Health literacy	Knowledge about signs or symptoms	1.3	2	
		Knowledge of therapeutic plan	1.4		
	Lifestyle	Lifestyle	1.8	1	
	Health status	Multimorbidity	1.9	2	
		Mental health	1.11		
Contextual	Healthcare access	Coordination of services	2.1	1	0.753
		Adequacy of healthcare services	2.2 2.3	2	
		Relation between patient and practitioner	2.4 2.5 2.6 2.7	4	
		The complexity of healthcare services	2.8 2.9	2	
		Healthcare professionals’ behaviour	2.10	1	
		Early disease diagnostic	2.11	1	
		Communication between patient and practitioner	2.12	1	
	Environment characteristics	Cultural barriers	2.13	1	

**Table 6 ijerph-19-03138-t006:** Construct Validity.

		Manage Disease in General Scale	Mental Health Inventory	Oslo Social Support Scale	Health Literacy Index
Individual	Pearson	−0.397	−0.386	−0.013	0.028
	Sig	<0.001	<0.001	0.891	0.823
Contextual	Pearson	0.360	0.231	0.201	0.453
	Sig	<0.001	0.023	0.044	<0.001
Score Health Management (Ind)	Pearson	−0.147	NA	NA	NA
	Sig	0.113	NA	NA	NA
Score Health System (Ind)	Pearson	NA	−0.330	NA	NA
	Sig	NA	<0.001	NA	NA
Score Social Support (Ind)	Pearson	NA	NA	−0.044	NA
	Sig	NA	NA	0.639	NA
Score Health literacy (Ind)	Pearson	NA	NA	NA	−0.206
	Sig	NA	NA	NA	0.076

## Data Availability

Not applicable. The data presented in this study are available on request from the corresponding author.

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
