# Peer review of "Patients’ Perspectives on Determinants Avoidable Hospitalizations: Development and Validation of a Questionnaire"

_ijerph, 2022, doi:10.3390/ijerph19053138_

Round 1
Reviewer 1 Report
This is an interesting work on an aspect of health care management that has a major impact on the quality and cost of services, namely inappropriate hospitalization for outpatient medical conditions. The article presents the study that creates and validates a new questionnaire for investigating patients' perspectives on the causes of these hospitalizations. The idea is original and there are no other similar instruments in the literature. Generally speaking, the study is structured and presented in a formally correct manner. However, there are improvements that the authors could make to the article.
1. I would recommend considering rewording the title, to increase the immediate comprehensibility of the context and type of study.
2. I would consider reducing the length of the Introduction, which is very informative but somewhat redundant.
3. Section 2.1, Phase 1, Literature review, more information is needed about the review process, e.g. when it was conducted, which databases were consulted, what publication interval was considered, keywords, etc.
4. I would consider whether it is possible to move some information from the Results to the Methods, in order to lighten the Results section.
5. Revision of English and correction of various typos in the text is recommended.
Author Response
- I would recommend considering rewording the title, to increase the immediate comprehensibility of the context and type of study.
We have changed the title to “Patients’ perspectives on determinants avoidable hospitalizations: development and validation of a questionnaire.
- I would consider reducing the length of the Introduction, which is very informative but somewhat redundant.
We have reduced the introduction.
- Section 2.1, Phase 1, Literature review, more information is needed about the review process, e.g. when it was conducted, which databases were consulted, what publication interval was considered, keywords, etc.
We have changed the Section 2.1, Phase 1, Literature review to clarify the review process. This review was not systematic, though it aimed at following the best practices in developing and validating scales for health research. The period it was conducted have been added.
- I would consider whether it is possible to move some information from the Results to the Methods, in order to lighten the Results section.
We have moved some information from the Results to the Methods section.
- Revision of English and correction of various typos in the text is recommended.
The manuscript has now been proofread.

Reviewer 2 Report
Thank you for giving me the opportunity to review this manuscript and congratulations to the authors for building such an interesting tool. From the point of view of the introduction / justification I have nothing to point out. Regarding the methodology, I believe that it is thoroughly described and that it adjusts to the stated objectives. However, I think that the authors should justify the reason for the sample size and sampling in phase 4. Although the authors honestly describe the weaknesses of the study, I think they need to explain better why they have not considered other habitual and determining variables, such as consumption of alcohol or tobacco, the level of health literacy, etc. If these variables have not been considered, on what type of patients can this questionnaire be used?
I consider that the questionnaire contains too many singularities and it will be difficult to apply it with guarantees outside of its original context. I think that the authors should explain in what contexts it could be used and I suggest adjusting the title of the manuscript to this circumstance.
In short, I believe that the biggest problem with this instrument is that it is difficult to extrapolate to other populations, although in the context in which it has been carried out it is probably very useful.
Author Response
Thank you for giving me the opportunity to review this manuscript and congratulations to the authors for building such an interesting tool. From the point of view of the introduction / justification I have nothing to point out. Regarding the methodology, I believe that it is thoroughly described and that it adjusts to the stated objectives. However, I think that the authors should justify the reason for the sample size and sampling in phase 4.
The authors appreciated the words of the reviewer #2 regarding the relevance of this tool that we also consider that could be useful to address the burden of admissions for ambulatory care sensitive conditions through considering the patients perspective.
The sample size has been justified in this manuscript version, and the number of participants involved in this study has also been discussed in the limitations (last paragraph of the discussion section).
Although the authors honestly describe the weaknesses of the study, I think they need to explain better why they have not considered other habitual and determining variables, such as consumption of alcohol or tobacco, the level of health literacy, etc. If these variables have not been considered, on what type of patients can this questionnaire be used? I consider that the questionnaire contains too many singularities and it will be difficult to apply it with guarantees outside of its original context. I think that the authors should explain in what contexts it could be used and I suggest adjusting the title of the manuscript to this circumstance.
In short, I believe that the biggest problem with this instrument is that it is difficult to extrapolate to other populations, although in the context in which it has been carried out it is probably very useful.
This is a relevant comment. The diseases often considered ACSC have different clinical manifestations and risk factors, thus including consumption of alcohol and tobacco in the questionnaire would not be fully adequate for the different diseases. Similarly, the questionnaire did not include health literacy or other specific determinants, as the aim of the questionnaire is to be comprehensive to different diseases and contexts, but also easy to use.
The questionnaire is a valuable tool that can provide a better understanding on individual and contextual factors associated to an avoidable hospitalization. It can have two important uses: first, when applied to patients that were hospitalized for conditions deemed avoidable, it can help identify shortcomings in the quality of the health care provided. Secondly, it can also help identifying those patients that present characteristics associated with higher risk of being hospitalized for ASCS, thus if indicates patients who should have be followed up more closely at the outpatient setting, to avoid rehospitalizations.
We have added this discussion in the final sections of the manuscript.

Round 2
Reviewer 2 Report
Thank you very much for the changes made. In my opinion the manuscript can be published. However I think it would be more appropriate to highlight the sociodemographic limitations of the questionnaire. In any case, thanks for addressing such an interesting topic and for building and validating the tool.
Author Response
Thank you very much for the changes made. In my opinion the manuscript can be published. However I think it would be more appropriate to highlight the sociodemographic limitations of the questionnaire. In any case, thanks for addressing such an interesting topic and for building and validating the tool.
Thank you for this comment. We have added an acknowledgement of the sociodemographic limitations of the questionnaire in the end of the discussion section; but also acknowledging that this questionnaire could provide a base for further language/context validations.